# Potential Targeting Mechanisms for Bone-Directed Therapies

**DOI:** 10.3390/ijms25158339

**Published:** 2024-07-30

**Authors:** Betul Celik, Andrés Felipe Leal, Shunji Tomatsu

**Affiliations:** 1Department of Biological Sciences, University of Delaware, Newark, DE 19716, USA; duzleme@udel.edu; 2Nemours Children’s Health, 1600 Rockland Rd., Wilmington, DE 19803, USA; andres.lealbohorquez@nemours.org; 3Institute for the Study of Inborn Errors of Metabolism, Faculty of Science, Pontificia Universidad Javeriana, Bogotá 110231, Colombia; 4Department of Pediatrics, Graduate School of Medicine, Gifu University, Gifu 501-1193, Japan; 5Department of Pediatrics, Thomas Jefferson University, Philadelphia, PA 19144, USA

**Keywords:** bone targeting, receptor mediation, glycobiology, bone and cartilage development

## Abstract

Bone development is characterized by complex regulation mechanisms, including signal transduction and transcription factor-related pathways, glycobiological processes, cellular interactions, transportation mechanisms, and, importantly, chemical formation resulting from hydroxyapatite. Any abnormal regulation in the bone development processes causes skeletal system-related problems. To some extent, the avascularity of cartilage and bone makes drug delivery more challenging than that of soft tissues. Recent studies have implemented many novel bone-targeting approaches to overcome drawbacks. However, none of these strategies fully corrects skeletal dysfunction, particularly in growth plate-related ones. Although direct recombinant enzymes (e.g., Vimizim for Morquio, Cerezyme for Gaucher, Elaprase for Hunter, Mepsevii for Sly diseases) or hormone infusions (estrogen for osteoporosis and osteoarthritis), traditional gene delivery (e.g., direct infusion of viral or non-viral vectors with no modifications on capsid, envelope, or nanoparticles), and cell therapy strategies (healthy bone marrow or hematopoietic stem cell transplantation) partially improve bone lesions, novel delivery methods must be addressed regarding target specificity, less immunogenicity, and duration in circulation. In addition to improvements in bone delivery, potential regulation of bone development mechanisms involving receptor-regulated pathways has also been utilized. Targeted drug delivery using organic and inorganic compounds is a promising approach in mostly preclinical settings and future clinical translation. This review comprehensively summarizes the current bone-targeting strategies based on bone structure and remodeling concepts while emphasizing potential approaches for future bone-targeting systems.

## 1. Introduction

Bone is an active metabolic tissue comprising inorganic and organic compounds along with several cell lineages [1]. Although bone remodeling is a dynamic process that ensures bone homeostasis in healthy individuals, bone-affecting diseases frequently impair this balance, leading to pathophysiological events, including inflammation, oxidative stress, and cell death [1,2,3], resulting in bone physiology impairment. Some bone-affecting diseases such as mucopolysaccharidoses (MPSs) [4], achondroplasia [5], osteogenesis imperfecta [6], thanatophoric dysplasia [7], among others, commonly lead to skeletal dysplasia. Unfortunately, when avascular zones in the bone are affected, the treatment becomes challenging [8]. For instance, in MPS IVA, a skeletal dysplasia caused by a lack of lysosomal hydrolase, the only Food and Drug Administration (FDA)-approved drug is based on the intravenous administration of the missing enzyme, which has a poor impact on the bone lesions [9]. The limited impact of this strategy has been mainly attributed to the low bioavailability of the enzyme in the avascular zones of the bone, such as the growth plate [10]. Growth plate chondrocytes are primarily affected in MPS IVA [4,11] (Figure 1).

Some efforts have been made to increase the bioavailability of several drugs in the bone [8]. For instance, a study conducted by Almeciga-Diaz et al. showed that aspartate octapeptides (D8) significantly increase the adeno-associated virus (AAV) affinity by the hydroxyapatite [13], the most abundant inorganic material in the bone [14], supporting that bone-targeting therapies could be an interesting alternative for the treatment of bone-affecting diseases. Likewise, drugs targeting bone cell receptors, such as receptor activator for nuclear factor-κB ligand (RANKL) [15], sclerostin [16], and the type 1 parathyroid hormone receptor (PTH1R) [17], have also shown promising outcomes in vitro and in vivo in target therapies. 

This paper comprehensively reviews the most relevant characteristics of the bone and its physiology and discusses how these insights can be turned into novel alternatives for bone-affecting diseases. A total of 188 original papers published in the last 10 years on PubMed, ScienceDirect, Google Scholar, and Web of Science were consulted. The search was conducted by using the following keywords and Boolean operators: (bone organic molecules AND bone-targeting) OR (bone inorganic molecules AND bone-targeting).

These searches helped us answer our research questions as follows:What are the potential molecules/compounds that can be utilized to target bone and cartilage?-We categorized them under organic and inorganic molecules.How do these molecules work? How are their structures modified?-We explained them briefly but not in detail to show their potential under current strategies.What are the roles of these molecules in cellular and bone development processes?-We provided information on their effects on the regulation of cellular processes.How are these molecules being implemented in preclinical/clinical studies?-A short summary of each study has been provided under each subtitle.What are the best potential molecules to target, specifically, the cartilage and bone growth plate?-Our specific goal is to find the best potential approaches for targeting bone growth plates and stimulating bone growth. Based on our particular interest, we focused on the molecules that can be utilized for growth plate targeting in the future, although we explained other targets in the current literature. Therefore, readers can find valuable information from an extended perspective. 

## 2. Organic Molecules in Bone Targeting

ECM is one of the organic components containing, especially, collagen proteins: the collagen types I, III, IV, V, and VI, and non-collagen proteins: proteoglycans, glycoproteins, and gamma-carboxyglutamate (Gla) domain-containing proteins.

### 2.1. Collagens

The most prevalent collagen in bone is type I, composing 95% of the collagen content, although other types are also present. Type I collagen has a critical role in forming collagen fibrils, which, in turn, interact with other collagens and non-collagen proteins to assemble the fibril bundles and fibers. The collagen fibrils are organized concentrically into lamellae of bone, and within the same lamellae, all collagen fibers are parallel to each other while they might be oriented at an angle of up to 90° relative to each other. These concepts provide tensile strength, elasticity, and rigidity to bone [18]. Based on their abundance and roles in bone and cartilage, collagens are the highlighted targets for drug delivery to the bone. Collagen binding in mammals occurs via discoidin domain receptors (DDRs) that are transmembrane proteins and a subclass of receptor tyrosine kinases (RTKs). The DDR family has two members: DDR-1, highly abundant in epithelial cells and leukocytes [19], and DDR-2, predominantly from mesenchymal origin, skeletal and heart muscles, kidney, and lung [20], which are the only RTKs interacting with structural components of the ECM and are not activated by soluble growth factors (Figure 2). DDR-1 is composed of five splice-variant isoforms (DDR-1a–e). At the same time, DDR-2 is alone but shares highly conserved sequences with the DDR-1 family. The DDR family consists of an extracellular ligand-binding domain, a transmembrane domain, and an intracellular region containing a kinase domain. The signal transduction is activated with the receptor-specific ligand binding. When collagen binds to the DDRs, DDR phosphorylation is stimulated and thus activates kinase activity, signaling to mitogen-activated protein kinase (MAPK), integrin, transforming growth factor-β (TGFβ), insulin receptors, and notch signaling pathways. Each DDR is stimulated by different collagens: DDR-1 by collagens I, IV, V, VI, and VIII and DDR-2 exclusively by the fibrillar collagen types I, III, and X [21]. Furthermore, DDR-1 is also activated by recognizing the GVMGVO peptide motif in both fibrillar collagens I, II, III, and VIII and non-fibrillar basement membrane collagen IV [21,22]. Collagen X is prevalent in the hypertrophic zone of the growth plate, which was shown to interact with chondrocytes, primarily with integrin α2β1. Later, the affinity of collagen X ligands was detected to be much higher for DDR-2 in the hypertrophic zone, and DDR-2 recognized the collagen X triple-helical binding site during receptor activation. The expression of DDR-2 was confirmed in the epiphyseal plate by RT-PCR and IHC [22]. Moreover, DDR-1 KO mice were characterized by the secondary ossification center’s delay in development and growth plate length increasing in the hind limbs. The proliferation of chondrocytes of the tibial growth plate was reduced, and Ihh, matrix metalloprotease (MMP-13), and collagen X were downregulated in chondrocytes, resulting in reduced terminal differentiation in the hypertrophic zone. As a result, decreased apoptosis delayed endochondral ossification and resulted in short stature and DDR-1-regulated terminal chondrocyte differentiation through the Ihh-Gli1-Gli2-collagen X pathway (Gli: effector transcriptional factor of Hedgehog signaling) [23]. Another study indicated that the silencing of DDR-2 contributed to severe skeletal growth and developmental defects in humans and mice. Expression and lineage analysis of chondrocytes showed selective expression of DDR-2 at the initial stages of bone formation in the resting zone and proliferating chondrocytes as well as periosteum. DDR-2+ cells could differentiate into hypertrophic chondrocytes, osteoblasts, and osteocytes and demonstrated a high level of colocalization with the skeletal progenitor marker Gli1. Ultimately, silencing of DDR-1 inhibited chondrogenic and osteogenic differentiation in limb bud chondroprogenitors and purified marrow-derived skeletal progenitors, respectively [24]. In MSCs, integrins α2β1 and α11β1 receptors interact with collagen I, ensuring cell survival and osteogenic differentiation via the protein kinase B (PKB/Akt) pathway. In contrast, the integrin α5 activation regulates human BM MSC (hBM MSC) differentiation into osteoblasts by activating focal adhesion kinase (FAK)/extracellular receptor kinase-1 (ERK-1/2)/MAPK and phosphoinositide 3-kinase (PI3K) signaling pathways [25,26]. DDRs modulate the differentiation of progenitor cells. The absence of DDR-1 in human adipose-derived MSCs reduces the chondrogenic genes and cartilaginous matrix deposition, suppressing the chondrogenic differentiation [27]. In the presence of collagen type I, hBM MSCs increased the expression of DDR-2 and α11β1. However, α11β1 was proven to be expressed significantly more during chondrogenic differentiation, while DDR-2 was in osteogenic differentiation [26,28]. The colocalization of DDR-2 was determined to correlate significantly with the expression of alkaline phosphatase (ALP) during mineralization [29]. Taken together, DDRs are one of the key factors in targeting bone-related cells. Particularly, the expression of DDR-2 on early-stage chondrocytes, which differentiate into trilineage cells, is an appealing approach for receptor-mediated endocytosis-related drug delivery strategies. Cartilage is a difficult-to-reach tissue, and the mechanisms underlying its uptake and signaling are still under investigation. However, as described in the literature for C-type natriuretic peptide (CNP) promoting bone growth [30], an administration of DDR-2-targeting peptide or reagents may provide options for the treatment of bone and cartilage diseases. 

### 2.2. Proteoglycans

Proteoglycans (PGs) comprise one or more glycosaminoglycan (GAG) residues covalently bound to the protein core. GAG residues in these cores include keratan sulfate (KS), chondroitin sulfate (CS), heparan sulfate (HS), hyaluronic acid (HA), and dermatan sulfate (DS). Sulfated negatively charged GAGs have varying roles in cell-cell and cell-matrix interactions due to their biological structures comprising a repeating disaccharide unit, uronic acid, and hexosamines [34]. PGs are commonly found in cartilage matrix mediated by endochondral ossification, indicating that the initial bone formation matrix is initiated by chondrogenic proteins already incorporated into this matrix. Moreover, matrix maintenance, organization, and regulation of cartilage calcification are modulated by PGs via the interactions with the GAG residues present in type II and IX collagen fibrils. Therefore, PGs and GAGs can inhibit hydroxyapatite formation and growth due to changing pH, physiological ionic strengths, large hydrodynamic size, and high charge density of these macromolecules [35]. Moreover, hydroxyapatite formation was detected in positive and negative surfaces and different conditions of supersaturated solutions [36]. GAGs, as pharmacological targets, have been analyzed primarily on carcinogenesis, metastasis, bone and joint, hematologic, and brain disorders [36,37]. Yang et al. (2015) studied the adhesive effect of CS to enhance the integration of bioglass (BG) in the treatment of bone defects in rabbit models. After the encapsulation of BM in the BG-CS composite (BG-CS-BM), rabbits with critical-size distal femoral bone defects were administered and sacrificed 4 or 6 weeks after treatment. The results showed significant bone growth in the BG-CS-BM-treated group compared to BG only and the empty control. BG-CS showed a similar effect in bone formation, although relatively lower than BG-CS-BM over 6 weeks of implantation. Nevertheless, BG particle migration after treatment remained an unmet challenge, triggering inflammation [38]. In another study, poly(L-lactide)-sucrose acetate isobutyrate (PLLA-SAIB)-coated collagen-CS biomaterial encapsulating rhBMP-2 was assessed in the bone defects in rats, and the results indicated optimum osteogenic induction as well as the biocompatibility of biomaterial [39]. rhBMP-9, in another study, was combined with HA to assess the adsorption and release in ST2 preosteoblasts. Following treatment, osteoblast differentiation was evaluated through ALP activity, alizarin red staining, and real-time PCR of osteoblast differentiation markers (ALP, COL1α2, and OCN). HA-BMP-9 treatment was shown to induce osteoblast differentiation highly, while a negligible effect was observed on cell proliferation. Moreover, HA alone induced osteoblast differentiation, which was 4-fold lower than with the A-BMP-9 combination [40]. On the other hand, KS has been investigated to target various diseases, such as cartilage damage and amyotrophic lateral sclerosis, inflammation, and malignant lymphoma [41]. KS is a critical GAG for maintaining a proper hydration level in skeletal tissues and resisting mechanical stress. KS is a highly accumulated GAG in bone and cartilage of mucopolysaccharidosis IVA (MPS IVA) patients due to the deficiency of N-acetylgalactosamine-6-sulfatase (GALNS) enzyme, resulting in the impairment of the further steps of KS catabolism. Hence, abnormal KS and C6-S levels appear in the tissues. Enzyme replacement therapy targets the GAGs accumulated. Although ERT slightly reduces this storage material in bone, disease progression is not fully corrected [42,43]. 

### 2.3. Small Leucine-Rich Proteoglycans

Small leucine-rich proteoglycans (SLRPs), for example, biglycan, decorin, keratocan, and asporin, regulate normal and pathological cellular behaviors, while fibromodulin, another type of SLRPs, regulates collagen fibrillogenesis. SLRPs participate in cell proliferation, osteogenesis, mineral deposition, and bone remodeling and thus contribute to maintaining bone homeostasis.

#### 2.3.1. Biglycan and Decorin

Biglycan and decorin are significant SLRPs in bone having CS/DS GAG substitutions. Biglycan expression begins in cell proliferation and plays a critical role in bone matrix deposition and mineralization, while decorin is continuously expressed starting from early bone matrix deposition [44]. Biglycan has close contact with some ECM proteins, such as collagen types I–IV and elastin, that provide stability and organization in tissues, and it is involved in hydroxyapatite formation in bone. Although biglycan is strongly connected to the ECM, some factors, including tissue stress and injuries, impair its bond with the ECM, and these unexpected circumstances cause proteases to release the PGs from ECM [45,46]. Decorin has a high affinity for collagen fibrils and regulates collagen fibril assembly through its DS chains. It forms a ring-mesh-like structure to bundle collagen fibers in the same direction in the ECM [47]. It simultaneously interacts with many RTKs, including EGFR, VEGFR-2, and mesenchymal–epidermal transition factor (Met) receptors, which leads to the inhibition of these RTKs (decorin as a pan-RTK inhibitor) (Figure 3) [46]. Besides its role in collagen fibril cross-linking and receptor mediation, decorin increases focal adhesion and proliferation of osteoblasts [48]. Osteoadherin is another SLRP containing the KS chain secreted by osteoblasts. It was found in the bovine mineralized bone matrix as a binding substance to hydroxyapatite [49].

#### 2.3.2. Keratocan and Asporin

Keratocan is expressed in osteoblasts and regulates bone formation and mineral deposition rates in the murine KERA gene KO model by modulating osteoblast function [49]. Asporin binds to collagen type I to promote collagen mineralization via the poly aspartate domain binding to calcium and regulating hydroxyapatite formation. However, the binding of asporin to collagen was inhibited by recombinant asporin fragment leucine-rich repeat (LRR) 10-12 and by full-length decorin. Asporin and decorin were shown to compete for binding to collagen. The presence of asporin triggers the production of collagen nodules and increases the mRNA of osteoblastic markers Osterix and Runx2 [50]. 

#### 2.3.3. Fibromodulin

Fibromodulin, a KS-chain SLRP, was found to be expressed by both chondrocytes and osteoblasts during fetal bone development. Moreover, it has interacted with the complement system and immune cells while functioning in inflammation-related diseases, such as osteoarthritis (OA), tendinopathy, atherosclerosis, and heart failure [51,52]. Fibromodulin and lumican, another KS-chain SLRP, were primarily determined in the precementum and pericementocyte area, and it was thought that they might have regulatory roles in the mineralization of cementum. Versican, biglycan, and decorin, DS-chain PGs, were also investigated for cementogenesis, and the results indicated that versican is found in the lacunae located in cementum and alveolar bone, decorin in the periodontal ligament and slightly in cementum matrix, and biglycan in cementoblasts/precementum.

In addition to classifying SLRPs via their N-terminal cysteine-rich clusters, evolutionary conservation, sequence homology, GAG chains, and biological functions, they can interact with various cell surface receptors, particularly RTKs and Toll-like receptors (TLRs) (Figure 3). Therefore, downstream signaling pathways are activated. In the case of tissue stress or injury, SLRPs are released to the sites to act as damage-associated molecular patterns [46,53]. Taken all together, GAGs and SLRPs are potential biomarkers and therapeutic targets in many bone diseases, including osteo- and peripheral chondromas, OA, bone-affecting lysosomal storage disorders, etc., and they regulate bone morphogenesis and homeostasis. Furthermore, HS membrane-binding proteoglycans, syndecans (SDCs), and glypicans (GPCs) have been shown to activate different signaling pathways in MSCs. Silencing of SDC-1 led to the inhibition of canonical Wnt signaling due to the deficiency of β-catenin in mice and triggered a pro-adipogenic phenotype with enhanced osteoblast maturation in MSC cultures. In contrast, the upregulation of SDC-2 reduced osteoblastic and osteoclastic precursors in the BM of mice, as well as Wnt/β-catenin signaling in osteoblasts. SDC-3 knockdown caused a low bone volume and deteriorated bone architecture in adult mice, which is promoted through canonical Wnt signaling in osteoblasts [24,26]. In the case of GPCs, the dysregulation of GPCs in hBM MSCs of OA patients decreased the protein level of extracellular negative regulators of Wnt/β-catenin during osteogenic differentiation [26]. Additionally, Ihh signaling was hyperactivated in GPC3-null mouse embryos, while GPC6 promotes the growth of developing long bones via Ihh signaling [54]. Figure 3Soluble SLRP-mediated downstream signaling has critical roles in cellular processes [46,55]. We summarize some of the signaling pathways; however, this diagram is still not fully completed because some receptors, such as TLR-2/4, have cross-talk mechanisms or interaction with different proteins (e.g., some cluster differentiation (CD) molecules; CD14 and CD44) in interleukin secretion, different pathways of angiogenesis (NOX-1/4 and ROS pathway), and autophagy. TGFβ and BMP are the receptors to which most SLRPs have different ways to bind and promote downstream signaling. The picture depicts biglycan binding to TGFβ and BMP to activate the SMAD pathways. Thus, the osteoblast-related genes can be transcribed for the bone formation process. CXCR is a receptor for keratocan that stimulates neutrophil granulation and activation. Angiogenesis is a critical process, particularly in bone development. Biglycan binding to TLR-2 and decorin binding to VEGFR promote angiogenesis by activating relevant downstream signaling. Additionally, biglycan binding to EGFR can trigger endosomal uptake mechanisms and further processes. This figure was created with BioRender.com.
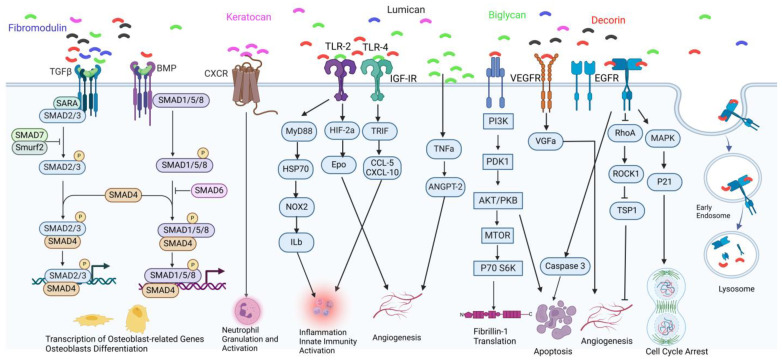


### 2.4. Glycoproteins

Glycoproteins contain a covalent linkage of sugar residues attached by asparaginyl or serinyl residue during the posttranslational modification process. These proteins in bone matrix enhance cell binding and, thus, they are used in various scaffolds for tissue engineering applications. The glycoproteins discussed here are ALP, osteonectin (ON), tetranectin, RGD-containing glycoproteins, thrombospondin, fibronectin (FN), vitonectin (VN), osteopontin (OPN), bone sialoproteins (BSPs), dentin matrix proteins (DMPs), and dentin sialophoshoproteins (DSPPs) [35,56]. 

#### 2.4.1. ALP

ALP is a prevalent membrane-bound glycoprotein in the placental, germinal, and intestinal tissues as tissue-specific ALP and in the liver, bone, and kidney as tissue-non-specific ALP. In bone, ALP released from either the osteogenic cell surface or matrix vesicles has key roles in the initial stages of hydroxyapatite formation, the deficiency of which causes hypophosphatasia (HPP), namely abnormal mineral deposition. ALP hydrolyzes inorganic pyrophosphate (PPi), increasing the level of inorganic phosphate (Pi) to initiate bone mineralization and adjusting the ratio of PPi to Pi. The sufficient accumulation of calcium ions and Pi within the matrix vesicles forms amorphous calcium phosphate or hydroxyapatite crystals, indicating the initial stage of ECM mineralization. Increased activity of ALP during osteogenesis reveals the differentiation of MSCs into osteoblasts. ALP is controlled by many mechanisms, including signaling pathways (Wnt, BMP-2, FGF, IGFBP/IGF) [57,58]. ALP has been entrapped within a hydrogel consisting of silk fibroin (SF) and a phosphorylated peptide (NapGFFY (NY)) and infused into a rat model. As a result, the mineralized SF gel promoted the osteogenic differentiation of rat BMSCs and regenerated the defective area of the femur in rats [59]. The deficiency of ALP contributes to a life-threatening perinatal or infantile case resulting from hypomineralization of the skeleton and dental areas. KO ALP mice were treated with AAV8 encoding TNAP-D10 within 5 days postnatally. Plasma ALP activity increased, and the accumulation of PPi was suppressed following treatment. Also, the longevity of treated mice extended compared to control groups. Bone microstructure with a few cortical defects and trabecular defects was similar to the wild type. On top of that, no adverse effects were detected in the kidneys, aorta, coronary arteries, or brain in 70 days postnatally, and there was no sign of rickets in the bowing of long bones, growing epiphyses, or fractures [60]. Human induced pluripotent stem cells (iPSCs) expressing ALP were evaluated in the treatment of HPP, which indicated that the ALP enzyme activity was enormously elevated in osteoblasts differentiated from these stem cells in vitro, and this was further confirmed with the increased expression of osteoblast biomarkers COL1a1 and OCN [61]. In clinical applications, asfotase alpha (Strensig®, Alexion, AstraZeneca, Boston, MA, USA) is a first-line approved enzyme replacement therapy for all patients diagnosed with HPP. Asfotase alpha comprises a catalytic domain of human recombinant TNSALP, Fc region of human immunoglobulin G1, and a deca-aspartate peptide (rTNSALP-Fc-deca-aspartate). This deca-aspartate peptide has a high affinity for binding hydroxyapatite. Therefore, the rTNSALP enzyme can be anchored to bone. In the clinical trial NCT02306720, patients who were administered with asfotase alpha for more than 6 months showed improvement in mobility, physical function, and quality of life during 3 years of follow-ups (Clinicaltrials.gov, accessed on 20 April 2024) [62].

#### 2.4.2. Osteonectin 

ON containing calcium-binding domains regulates the bone mineralization process, cell cycle, cell-matrix interactions, and cell morphology [NO_PRINTED_FORM]. Besides calcium binding, ON can bind to different collagens (I, III, and V), thrombospondin, and hydroxyapatite via calcium-binding sites with high affinity. ON has been thought to regulate the growth and proliferation of mineral crystals since it can accumulate only in the mineralized matrix and not initiate mineralization, while it can stimulate angiogenesis and the production of MMPs. ON can be induced by growth factors (TGF-b and FGF-2), BMP-2, and glucocorticoids [35,63]. The decreased rate of bone formation in ON-null mice has been related to osteopenia with significant loss of trabecular bone. Moreover, these mice illustrated a reduction in osteoblast and osteoclast surface markers and numbers, indicating decreased bone turnover. The data suggested that a progressive decrease in the number of cells in bone led to osteopenia in ON-null mice, contributing to bone fragility. ON has critical roles in bone remodeling and maintenance of bone mass, which may play roles in inherited susceptibility to OP [63]. Recent studies revealed that ON bidirectionally regulated the mineralization of osteoblasts with a dose-dependent regulatory effect and activated the p38 pathway via collagens binding to DDR-2 receptors (Figure 2) [64]. ON activates collagens to bind to DDR-2. With that, collagen synthesis is stimulated, and conformational changes of collagen are induced. Therefore, the activation of DDR-2 can regulate the interactions of cells and ECM via the p38 MAPK pathway. This activation can also trigger pathological changes in chondrocytes and osteoblasts, further resulting in bone formation via the p38 pathway. This pathway was considered as the critical signaling pathway for normal skeletal development [64,65,66]. 

#### 2.4.3. Tetranectin

Tetranectin, a calcium-binding protein from the C-type lectin family, is primarily found in serum and ECM during development, tissue regeneration, and cancer. It can bind to ECM components, including fibrin and plasminogen, stimulating proteolytic activity and growth factors and regulating ECM proteolysis. Normal adult tissue has low levels of tetranectin; however, it decreases in plasma in the case of pathological conditions involving cancer, inflammatory diseases, and coronary artery disease [67]. In bone, osteoblasts express tetranectin during mineralization. In previous studies, TGFβ was shown to downregulate dexamethasone-induced tetranectin gene expression in osteoblastic cells in vitro. Newborn mice showed tetranectin immune reaction in the newly formed woven bone around the cartilage and along the periosteum, while no activity was observed in the growth plate or surrounding skeletal muscle. Tetranectin was shown to have potential effects on heart failure as a biomarker expressed in the myocardium [67]. Although the biological function of tetranectin in osteogenesis and chondrogenesis is still unknown, tetranectin–plasminogen activation pathways were shown to inhibit periosteal chondrogenesis and promote periosteal osteogenesis during fracture healing under the administration of inhibitors for plasminogen activators. This inhibition also suppresses the differentiation of MSCs into osteoblasts in mice [68]. 

### 2.5. RGD-Containing Glycoproteins

Bone ECM contains some glycoproteins with RGD amino acid sequences recognized by cell surface markers called integrins. These RGD-containing glycoproteins promote the cell-matrix interactions. Thrombospondin, fibronectin, vitronectin, and a family of small integrin-binding ligand N-linked glycoproteins (SIBLINGs) comprising of osteopontin, BSPs, dentin matrix protein-1, dentin sialophosphoproteins and matrix extracellular phosphoglycoproteins are important biomarkers and each biomarker has been assessed in a variety of diseases, mostly in cancers, such as metastatic bone disease [69] and breast cancer [70], and infective diseases [71]. 

#### 2.5.1. Thrombospondins

Thrombospondins (TSP1–5) expressed during skeletal development have been shown to bind several matrix proteins, growth factors, and cell surface receptors. TSP4 regulates anti- and pro-angiogenic processes, and TSP1 and -2 act as inhibitors of angiogenesis and modulators of collagen fibrillogenesis in bone and tendon. Apart from that, TSP1 and -2 influence MSC fate and lineage progression, while TSP3, -4, and -5 affect ECM and chondrocyte organization in the growth plate [72]. Among all TSPs, TSP4 has been found to have critical roles in tissue remodeling, including the initial stage of osteogenesis, mature tenocytes, larval limb development, connective tissues, vasculature, and heart [73,74,75]. The deficiency of TSP4 in mice led to the disorganization of collagen fibers in the tendon and disruption in ECM [76]. Studies showed that TSP4 played an adaptor protein role in ECM assembly. Furthermore, it was upregulated in wound healing and hypertrophic scar formation, as well as collagen I. Thus, TSP4 might be a new target in osteogenic lineage-related diseases.

#### 2.5.2. Fibronectins

Fibronectins (1–3) are among the most prevalent ECM components in bone and many connective tissues. They promote cell adhesion, growth, migration, and differentiation based on the interactions between cells and ECM. Collagens, as stated above, are vital components of ECM, and fibronectin has a high affinity to collagens I and II, as well as binding to heparin and integrin. Besides the other two units binding to collagen collectively, the type 2 unit of fibronectin collagen-binding domain is present in mannose receptors and MMP-2. Interestingly, fibronectin binds to denatured collagens [77,78]. Type 2 fibronectins might be utilized to bind more collagens and provide an adhesive environment for cellular homeostasis. Furthermore, fibronectin matrix assembly is a key factor in mesenchymal cell condensation during chondrogenic development, and impairment of this assembly, along with other transportation genes, results in chondrodysplasias [79]. Overall, this approach might be beneficial during MSC transplantation. More studies are required to understand the efficiency of this approach. 

#### 2.5.3. Vitronectin

Another abundant glycoprotein found in serum, ECM, and bone is vitronectin, which promotes cell adhesion, spreading, and migration. A new vitronectin-derived peptide, VnP-16 (RVYFFKGKQYWE), was evaluated in bone loss induced by ovariectomy in mice, in which VnP-16 accelerated osteoblast differentiation and activity via β1 integrin receptors/focal adhesion kinase (FAK) signaling, thus promoting bone formation. VnP-16 was also shown to inhibit bone resorption by suppressing Janus N-terminal kinase (JNK)-c-Fos-NFATc1 (NFATc1; nuclear factor of activated T cells cytoplasmic 1) and αvβ3 integrin-c-Src-PYK2 (PYK; proline-rich tyrosine kinase-2) signaling. Therefore, osteoclast differentiation was negatively regulated. The anabolic effect of VnP-16 on osteoblast differentiation and expansion increased bone regeneration while significantly alleviating pro-inflammatory cytokine-induced bone resorption via blocking osteoclast differentiation and function in mice, which resulted in reversed ovariectomy-induced bone loss [80]. Additionally, human MSCs were shown to adhere to ECM proteins with varying affinity via distinct integrin receptors. Although fibronectin was considered to have the highest affinity (fibronectin, collagen I, collagen IV, vitronectin, and laminin 1, respectively), the osteogenic differentiation occurred in cells cultured on vitronectin and collagen I, not others. Here, vitronectin and collagen I induced osteogenic differentiation of human MSCs [81]. 

#### 2.5.4. Osteopontin

Osteopontin (OPN), a secreted phosphoprotein (SPP), is a SIBLINGs family member with an RGD sequence and modulates cell adhesion via autocrine and paracrine factors. Although OPN was considered only in ECM of bone, it was detected in dentin, cartilage, kidney, and vascular tissues with specific roles, such as acting as a structural molecule and a cytokine to mediate cellular communications. The RGD domain allows it to bind to αvβ1, αvβ3, αvβ5, α4β1, α5β1, and α9β1 integrins in many cell types: leukocytes, epithelial cells, smooth muscle cells, etc. [35,82]. In a study, OPN was analyzed regarding the origin, in which tumor-derived OPN was primarily soluble and did not support the proliferation of endothelial cells and prevent their apoptosis without adhesion. Hence, OPN might have different structures and functions in cells originating from other organs [83]. OPN has a high affinity to αvβ3 integrins and CD44 receptors in the attachment of osteoclasts to bone ECM, by which OPN regulates proliferation, differentiation, inflammation, metabolism, and tumor metastasis in chondrocytes, osteoblasts, osteoclast synoviocytes, and MSCs. It has roles in cartilage maintenance and subchondral bone homeostasis to regulate ECM, affecting bone turnover, bone mineral density (BMD), morphological formation, and reconstruction. The occurrence and development of many bone-related diseases, such as OP and OA, are associated with OPN. Additionally, an increase in OPN contributed to low BMD and OP, the level of which in serum has been correlated with the severity of OP and provided an early diagnosis of the disease [84]. OA is a complicated disorder involving excessive immune reactions. OPN is also a biomarker for OA affecting joints and surrounding tissues. It might regulate the degenerations in articular cartilage by modulating the expression of MMP-13, digesting type II collagen, and degrading GAGs. With the administration of OPN, OA-related cartilage inflammation decreased, which was found to be associated with the ERK1/2 pathway [85,86]. 

#### 2.5.5. Bone Sialoproteins

Bone sialoproteins (BSPs) accumulate in cement lines and fill the gaps between mineralized collagen fibrils, which are present in bone, dentin, cementum, and hypertrophic chondrocytes. The expression of BSPs is high in osteoblasts, osteoclasts, osteocytes, and chondrocytes [35]. They may have roles in mediating the early stage of matrix mineralization and the late stage of osteogenic differentiation, and they show high binding affinity to calcium. Furthermore, the level of BSPs influences the mineralization of the bone matrix. In addition, BSPs affect the proliferation and expansion of osteoblastic phenotype from BM. Overexpressed BSPs accelerate the matrix mineralization, whereas reduced levels result in poor mineralization. This can be explained by their binding affinity to type I collagen [87]. 

#### 2.5.6. Dentin Matrix Proteins

Dentin matrix proteins (DMPs) are a group of SIBLINGs that modulate cell attachment, proliferation, differentiation, and matrix mineralization of dentin and bone. Four types exist (DMP1, DMP2, dentin sialoprotein (DSP), and DMP4). DMP1 is a highly phosphorylated type with a strong affinity to bind calcium. Moreover, the N-telopeptide of the α1 chain of type I collagen carries the binding site of DMP1. DMP1 binds the gap regions presented in collagen fibrils, and N-telopeptide is in these regions. The binding of DMP1 to N-telopeptide in the gap regions facilitates the nucleation of hydroxyapatite and polymerization of collagen during tissue calcification. Namely, DMP1 is considered a nucleator of hydroxyapatite formation, indicating that DMP1 and collagen interaction might initiate the mineralization process [88]. The inactivation of DMP1 upregulates FGF receptor signaling in osteocytes, which results in excessive production of FGF23 in X-linked hypophosphatemic rickets (XLH) and autosomal recessive hypophosphatemic rickets type 1 in humans and mice. For instance, burosumab-twza (Crysvita®) neutralizing monoclonal antibodies (anti-FGF23) was the first approved treatment for adults and children with XLH (for the age of 1 year old and older) and tumor-induced osteomalacia [89,90].

#### 2.5.7. Dentin sialophosphoprotein

Dentin sialophosphoprotein (DSPP), described as enhancing nucleation and growth of hydroxyapatite, is cleaved into two proteins: dentin sialoprotein (DSP) and dentin phosphoprotein (DPP). In a study, DSPP-null mice reflected a similar phenotype of human autosomal dominant dentinogenesis imperfecta [91]. DPP alone was thought to initiate and maturate dentin mineralization. To confirm, DSPP-null mice were genetically modified to express DPP only (not DSP), and the conditional DPP KO mice were generated in that background. DPP expression partially corrected predentin width, and the phenotype was partially ameliorated. On top of that, irregular unmineralized areas in dentin disappeared. Although a significant recovery was detected in dentin volume, no change occurred in the dentin mineral density. In conclusion, DPP was found to be involved in the maturation of dentin mineralization, while DSP initiated the dentin mineralization [92].

### 2.6. Gla-Containing Proteins

Gla-containing proteins are posttranslationally modified by vitamin K-dependent enzymes, including osteocalcin (OCN) and matrix Gla protein. OCN, known as γ-carboxyglutamic acid-containing protein secreted by osteoblasts, is the most prevalent non-collagenous protein in bone (~25% of non-collagenous matrix protein) and regulates cartilage calcification. Carboxylation of OCN occurs in osteoblasts before their release into the bone ECM. However, both carboxylated and non-carboxylated forms of OCN can be present in the circulation, which is in close connection with energy and glucose metabolism [93], as well as bone metabolism to confirm osteoblastic activity as a biomarker. The carboxylation status of OCN affects its formation and structure in the skeleton and its function in the endocrine system regarding energy metabolism. Also, it regulates the stability and binding potential of OCN via its cellular G-protein coupled receptor-6A (GPCR6A) receptors. While tricarboxylated OCN has a function in the configuration of ECM, undecarboxylated ones regulate hormonal functions. Also, the carboxylation degree determines OCN’s interactions with Ca^2+^ [94]. Posttranslational modifications of OCN are regulated by vitamin K-dependent carboxylation of glutamate residues into highly Ca-binding Gla residues, and carboxylated, undecarboxylated, and undercarboxylated OCNs are circulated in the bloodstream. Hence, OCN directly regulates the insulin-responsive glucose transport system in myocytes and adipocytes and increases insulin sensitivity [95]. OCN is thought to have a high affinity to the mineral component of bone due to its acidic character, which binds to the calcium of hydroxyapatite to regulate the growth of mineral composition in bone. Its binding to hydroxyapatite accelerates the nucleation of the hydroxyapatite and functions in cell signaling and the recruitment of osteoclasts and osteoblasts. In MSCs, loss and rescue of OCN and OPN were found to modulate osteogenic and angiogenic features of these cells along with matrix integrity and mineral quality. The loss of OPN and OCN in MSCs of mice led to the involvement of ECM integrity and maturation as well as an unexpected increase in GAG composition associated with skeletal connective tissues, resulting in a delay in the maturation of mineral content. Treating these MSC cultures with OPN and OCN supplements rescued their proliferation, osteogenic potential, matrix integrity, and mineral quality [96]. Matrix Gla protein (MGP), a vitamin K-dependent protein, is more prevalent in cartilage than in bone, and its expression begins in the early stage of development and remains at the same level throughout life [35]. MGP undergoes two posttranslational modifications to function: phosphorylation and carboxylation, and all these isotypes are present in the circulation. The disruption of MGP function results in failure in the anti-calcification process in the vasculature, particularly resulting in coronary artery diseases [97]. This protein is highly expressed at the cellular level in MSCs, vascular smooth muscle cells (VSMCs), and chondrocytes [97]. MGP, a secreted extracellular protein, has a high binding affinity for calcium, phosphate ions, and hydroxyapatite. MGP is an essential target in low bone mass-related diseases, which regulates osteoclast differentiation and function because bone resorption is maintained by osteoclast activity [98]. The inhibition of calcification occurs in many ways: direct inhibition of calcium phosphate precipitation, the formation of matrix vesicles, the formation of apoptotic bodies, and trans-differentiation of VSMCs. Although their mechanisms of action remain unclear, they are considered involved in the functional inhibition of BMP-2 and BMP-4 since they block calcium crystal deposition and shield calcification [97]. Lastly, the silencing of MGP elevates intracellular Ca^2+^ flux [98].

### 2.7. Plasma Membrane Proteins

#### 2.7.1. Adhesion Molecules

Evidence suggests that chondrocytes interact with the ECM through adhesion molecules [99]. Among others, the integrin α1β1 is a well-known protein leading to chondrocyte and collagens II, IV, and VI, matrilin-1, as well as laminin-1 interactions [99,100]. Interestingly, integrin α1β1 is overexpressed during early OA, protecting against posttraumatic OA [101]. Similar findings have been described for β2 integrins [102]. For example, integrin β1 can be expressed in chondrocytes in different zones of bone, which triggers the inhibition of integrin-mediated signal transduction and thus stimulates chondrogenesis. However, the expression of this integrin in diseases like OA was significantly downregulated. In other words, integrin β1 functions as a protector against arthritis [103]. The expression or conformational regulations of different integrins in the cell surface are associated with pathological conditions. In OA, alterations in subchondral bone microarchitecture abnormally increase the αV integrin-mediated activation of TGFβ in articular cartilage chondrocytes. Furthermore, the severity of TGFβ activation is aligned with mechanical stress distribution. This excessive activation of TGFβ results in cartilage degeneration in OA mice. This activation can be suppressed by the natural binding of arginylglycylaspartic acid (RGD) peptide ligand. Overall, all these processes are stimulated by mechanical stress, and the increased severity of this stress strongly connects αV and RGD sequences to mediate excessive TGFβ activation via cell contractile forces [104].

#### 2.7.2. Ion Channels

Although chondrocytes are not excitable cells, like neurons, ion channels play critical roles in non-excitable cells to keep their homeostasis (Figure 4). Ion channels are pore-forming transmembrane proteins that regulate membrane potential maintenance through ion transport [105]. They are crucial for cell homeostasis, including resting potential maintenance and signal transduction, and can be classified into voltage- and ligand-gated channels [106,107]. The ionic microenvironment in cartilage is a cation-rich place primarily represented by sodium (Na^+^), calcium (Ca^2+^), and potassium (K^+^) due to the presence of large amounts of negatively charged proteoglycans [108,109]. Such ions are well-known to regulate cell proliferation and differentiation in chondrocytes. In fact, electrophysiological recording using whole-cell patch clamp experiments demonstrated that pharmacologically induced depolarization by blocking of voltage-gated potassium (Kv) via tetraethylammonium avoids cartilage synthesis in chondrogenic cells by impaired intracellular Ca^2+^ ([Ca^2+^]_i_) oscillation [108], suggesting pivotal roles of calcium-permeable channels.

Calcium channels have been implicated in several cartilage-related diseases. For instance, in rheumatoid arthritis (RA), a common autoimmune disease causing cartilage damage [110], the acid-sensitive channels [111], transient receptor potential channels [112], P2X7 receptors [113], and Piezo channels [114], among others, have been involved in RA pathogenesis by the regulation of calcium ion-related signaling pathways, which results in synovial hyperplasia, structural damage, and inflammatory burst [110]. Notably, the Piezo channels, which are permeable to monovalent (i.e., Na^+^, K^+^) and divalent (i.e., Mn^2+^, Ca^2+^) ions gated by mechanical forces [115], were initially reported in the plasma membrane of chondrocytes by Lee et al., 2014 [116]. In elegant experiments, the authors found that induced mechanical injury on articular chondrocytes leads to the Piezo channel activation, which seems to act in conjunction with L-type voltage-gated Ca^2+^ channels (VDCCs) for [Ca^2+^]_i_ mobilization. The L-type VDCC is well-known to regulate chondrocyte-related cell processes in the growth plate and cartilage regions [117,118]. Interestingly, mechanical injury-induced chondrocyte death was prevented by blocking Piezo channels with GsMTx4 [116], a spider venom peptide that inhibits cationic mechanosensitive channels [119]. Although those results suggested a pathological role of Piezo channels in the pathogenesis of RA through the induction of chondrocyte apoptosis [110], a recent study conducted by Hendrickx et al., 2020 revealed a dual role for these channels [120] since their lack caused severe early-onset OP with spontaneous fractures in mice as well as an impaired endochondral ossification [120]. Piezo channel activation was found to be strongly associated with BM MSC differentiation into osteoblasts; chondrocytes to promote apoptosis and endochondral ossification; osteoblasts to promote bone formation, to reduce proliferation, and inhibit bone resorption by osteoclasts; osteocytes to promote bone formation [121]. Although Piezo1 could be a potential target for cartilage diseases, its modulation should be performed carefully.

Another well-characterized calcium channel in chondrocytes is the large conductance Ca^2+^-activated K^+^ (BK_Ca_) channel, which is encoded by the KCa1.1 gene [109,122]. BK_Ca_ are channels activated by either membrane depolarization or [Ca^2+^]_i_ elevation, resulting in K^+^ efflux and membrane hyperpolarization [123]. Paxilline blocking of BK_Ca_ channels results in overexpression of RUNX2, a transcription factor involved in the development and maintenance of the bone, as well as an increased migration pattern of chondrogenic progenitor cells, suggesting that modulation of BK_Ca_ regulates cell properties in chondrogenic lineages [124]. Interestingly, BK_Ca_ channels have shown differential expression patterns in pathological states such as OA [124]. For instance, KCNMA1 is overexpressed in OA chondrocytes, and it has also been established that cytokines such as IL-1β and TNFβ lead to KCNMA1 overexpression in OA cartilage [125], suggesting that pro-inflammatory events could modulate BK_Ca_ channels.

Finally, early works showed that chondrocytes require glutamate for their proliferation in an N-methyl-d-aspartate (NMDA) receptor-dependent way [126]. Most recently, Matta et al. (2019) supported such initial observations and determined that NMDAR-mediated calcium signaling is pivotal for proliferation and ECM synthesis in early states [126]. Differential NMDA isoform expression patterns are detected in OA chondrocytes [127], and it has been observed that IL-1β leads to NMDA activation [128].

Despite the evident compromise of ion channels in cartilage-related therapies and their potential pharmacological modulation, they are poorly explored in severe chronic entities such as metabolic diseases affecting bone. Novel approaches to elucidate the expression pattern of ion channels will surely help to better understand the natural history of those diseases and attempt new treatments targeting such channels.

#### 2.7.3. G-Protein Coupled Receptors (GPCRs)

GPCRs are the largest receptors in eukaryotes and lead to the transduction of hundreds of signals [129]. GPCRs comprise a single polypeptide folded into a globular shape, forming seven segments on the plasma membrane with loops in the extracellular and cytoplasmatic faces [129]. Those receptors are attached to G-proteins, which bind nucleotides guanosine triphosphate (GTP) or guanosine diphosphate (GDP). Currently, four G-protein families are described [130]: Gs, which activates adenyl cyclase; Gi, which inhibits adenyl cyclase; Gq/11, which is involved in IP3 and diacylglycerol, intracellular calcium mobilization, and protein kinase C activation; G12/13, which is associated with rho GTPase.

GPCRs have been implied in several physiological events of chondrocytes. For instance, chondrocyte differentiation is partially attained through G-protein signaling modulation via the parathyroid hormone receptor (PPR) [131]. The PPR activates two stimulatory G-protein α-subunits, Gqα and G11α G-proteins [132]. In this regard, the Gαs seem to be a critical regulator of chondrocyte differentiation through their negative regulation. Mutations in the GNAS gene, which encodes for Gαs, leads to Albright hereditary osteodystrophy, a rare disease characterized by growth abnormalities due to the acceleration of chondrocyte differentiation and interruption of longitudinal bone growth in those patients [133,134]. Similar findings were also reported by Chagin et al., 2014 by using GNAS^−/−^ mice, in which authors showed severe growth retardation in the animals, as compared to wild type, due to an increase in chondrocyte differentiation and apoptosis but not in proliferation [133]. Gqα and G11α prevent chondrocyte apoptosis upon Gα ablation [131].

As with PPR, many other GPCRs such as prostaglandins [135], the RDC1 receptor [136], β-adrenergic receptors [137], and histamine H2 receptors [138], among others, have been described in chondrocytes and could have important roles in chondrocyte physiology, since they can also activate Gαs. Those receptors could be potential targets for treating diseases affecting chondrocytes. Indeed, it was also reported by Carlson et al., 2021 that pharmacological inhibition of GPCR kinase 2 via paroxetine, a classical anti-depressant, abated OA progression and promoted cartilage regeneration in an OA mouse model [139], supporting that GPCRs could be an interesting target for chondrocyte modulation. Several GPCR-modulating drugs are currently approved by the Food and Drug Administration [140].

#### 2.7.4. Wnt Signal and LRP 

The Wnt/β-catenin signaling pathway is a well-conserved pathway involved in bone formation and bone remodeling through the interaction of WNT molecules and the coreceptor LRP5/6, which leads to the β-catenin reaching the nucleus due to its association with the transcription factor Tcf/Lef to promote cell proliferation (Figure 5) [141,142]. LRPs are low-density lipoprotein receptors distributed in the plasma membrane of several cells, including chondrocytes [142]. ST and Dkk-1 (Wnt inhibitor proteins) can bind LRP 5/6 [143]. SOST expression has been reported in chondrocytes of the growth plate from biopsies of patients with OA [144]. The deficiency of LRP5 (loss of function) causes osteoporosis–pseudoglioma (OPPG), while the impaired SOST stimulates bone formation in mice and humans. The SOST gene encodes sclerostin (bone formation inhibitor) binding to LRP5/6 Wnt coreceptors, regulating bone formation. Thus, patients with OPPG resulting from loss of function in the LRP5 gene should be administered with an antagonist of SOST/sclerostin action, while the inhibitors of WNT signaling may hinder the overgrowth of bone in SOST deficiency-related sclerosing diseases [145]. Romosozumab, a humanized neutralizing monoclonal antibody (anti-sclerostin), is an approved treatment for postmenopausal women with osteoporosis. This anabolic treatment increases bone formation and reduces bone resorption. In the clinical trial NCT01575834, the romosozumab treatment lowered the risk of clinical fracture by 73% compared to the placebo group [146]. 

Several chondrocyte-related diseases are characterized by pro-inflammatory events [147]. Thus, potential receptor regulation is feasible. In this scenario, it was previously demonstrated that IL-1β increases the LPR5 expression in human and mouse OA chondrocytes, which potentiates the Wnt/β-catenin signaling and the synthesis of matrix metalloproteinase-13 (MMP-13) by promoting cartilage destruction [148]. Thus, LPR5 could be a potential target for treating bone-related therapies.

On the other hand, LRP1 is a chondrocyte receptor involved in the regulation of a disintegrin and metalloproteinase with thrombospondin motifs 5 (ADAMTS-5), the major aggrecan-degrading enzyme in cartilage, through their LRP-mediated endocytosis [149]. Interestingly, this process is dysregulated during OA, promoting the increased extracellular levels of ADAMTS-5, leading to the loss of aggrecan and collagen fibrils due to an apparent downregulation in the LRP1 translation [150]. LRP1 also binds ADAMTS-4 and -5 and MMP-13 [149,151]. Consequently, potential strategies targeting the restoration of LRP1 in the chondrocyte plasma membrane to physiological levels could propose novel alternatives for diseases such as OA. Figure 5Wnt signaling pathways in bone metabolic processes [152,153,154]. The figure shows that the canonical (**left** panel) and non-canonical (**right** panel) Wnt signaling pathways regulate different metabolic processes in bone, including bone development, skeletal homeostasis, and regulation of bone mineral density. In the canonical state, Wnt binds to LRP5/6 and FZD receptors to recruit dishevelled (DVL) and Axin that stabilize β-catenin cytoplasm. Then, nucleus-translocated β-catenin binds to transcriptional proteins for the transcription of related genes. This pathway cooperatively works with BMPs. β-catenin/TCF-4 binding transactivates BMP-2 transcription via TCF/LEF response element in the BMP-2 promoter. Wnt and BMP cascades are individual mechanisms by which bone formation and resorption are regulated. Wnt signaling regulates the differentiation of pluripotent MSCs into osteoprogenitor cells, while the BMP pathway stimulates these progenitor cells into mature osteoblasts. Afterward, Wnt signaling cooperates with BMP to promote osteoblast differentiation (bone formation), whereas osteoblast-derived Wnt-5a signaling activates the non-canonical cascade in osteoclasts to stimulate their differentiation (bone resorption). An increase in ALP activity and mineralization is a sign of this process [155,156]. On the other hand, the activation of Fzd/Ror/Ryk receptors via non-canonical Wnt binding recruits DVL. With that, heterotrimeric G-protein (Gq) activates PLC, breaking down phosphatidylinositol 4,5-bisphosphate (PIP2) into IP3 and DAG (second messengers). IP3 stimulates Ca^2+^ release signaling in ER and activates several mechanisms, while DAG promotes the activation of PKC. By depicting the non-canonical pathway, we emphasize that Ca^2+^ is an important component of the hypertrophic stage. Ca^2+^ is prepared via two mechanisms: extracellular Ca^2+^ influx and release from ER storage. When Ca^2+^ in cytoplasm binds to calmodulin (CaM), Ca^2+^/calmodulin-dependent protein kinase (CaMKII) is auto-phosphorylated. Therefore, hypertrophy of chondrocytes is derepressed by the activation of CaMKII phosphorylating histone deacetylase 4 (HDAC4). Another protein called 14-3-3 binds to HDCA4, blocks the translocation of HDCA4, and stimulates transportation of nuclear HDCA4 into cytoplasm. Hence, HDCA4 activity is suppressed, and hypertrophy-related genes in chondrocytes are transcriptionally activated. Furthermore, the histone deacetylation via HDCA4 prevents RUNX2 (one of the hypertrophic transcription factors) [152,153]. This figure was created with BioRender.com.
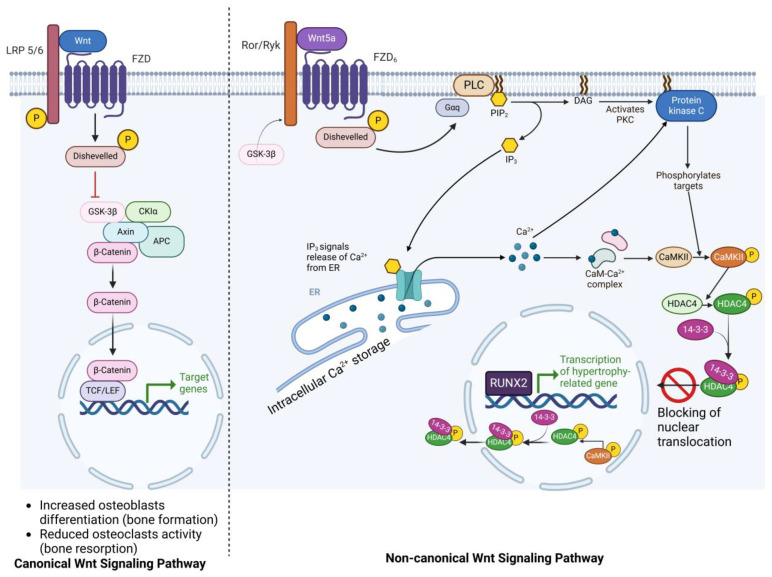


## 3. Inorganic Component of Bone and Targeted Drug Delivery

The major inorganic component of hard tissues, including bone and dentine, is hydroxyapatite, which occurs via biomineralization, in which minerals and ECM interact. The deposition of hydroxyapatite is provided by collagen, acting as a template for mineralization [56]. As a therapeutic target, hydroxyapatite is highlighted for drug targeting and has been widely evaluated in many clinical settings. 

### 3.1. Bisphosphonate

Bisphosphonates (BPs) have been investigated in many bone-related diseases, such as Paget’s disease, OP, metastatic bone disease, and various rare bone disorders, as well as in non-skeletal diseases. In bone, BPs have a high affinity to hydroxyapatite; however, this binding varies based on mineral deposition sites. In other words, newly deposited mineral sites in osteoid are most likely preferred by BPs, followed by resorption sites and bone turnover sites [157]. BPs are resistant to hydrolysis due to their pyrophosphate analogs, by which they gain anti-osteoclastic activity, thereby preserving bone mineral density. BPs inhibit calcification and hydroxyapatite breakdown, resulting in bone resorption suppression. Briefly, BPs contain two side chains on the phosphate–carbon–phosphate backbone: R1, where the affinity to bone is determined, and R2, where the potency of BPs is present. Nitrogen determines the potency of BPs, and the nitrogen side chains directly bind to the hydroxyl groups on the hydroxyapatite, increasing the affinity. Nitrogen-containing BPs (e.g., zoledronate, alendronate, etc.) are more potent than others (e.g., clodronate, etidronate, etc.). Nitrogen-containing R2 chains are positively charged, and binding them to the hydroxyapatite surface turns the surface charge into a more positive potential. These changes attract more negatively charged phosphonate groups and thus enhance the binding affinity. Furthermore, the structure of nitrogen in R2 influences the degree of anti-resorption activity. For instance, risedronate is the most potent due to the heterocyclic nitrogen ring [158,159]. 

The mechanism of action of BPs at the cellular level begins with their binding to osteoclasts (entering bone), which inhibits the prenylation of small GTPases and triggers apoptosis of osteoclasts while preventing the apoptosis of osteoblasts and osteocytes. Osteoclasts engulf BP in large quantities and non-resorbing cells in small quantities, which are released to become available for natural desorption [160,161]. The distribution of BPs is not homogeneous in the skeleton due to the surface area of cortical and trabecular bone, in which trabecular bone takes up more BPs resulting from high bone turnover. Locally increased bone turnover in metastatic bone diseases or Pagetic lesions contributed to high BP binding. To detect the binding of BPs, technetium-99m (99mTc) was bound to BP with an ionic bond. Bone scanning data indicate positive bone metastasis at lytic and blastic sites but were negative in lytic lesions from multiple myeloma. In these cases, BP binding has been considered to correlate with three phenomena: ALP activity, newly deposited mineral availability, and disassociation of bonds between 99mTc and BP at osteolytic sites due to the acidic environment [157]. In another study, measurement of tibia, rib, and vertebrae confirmed that the fluorescence-tagged low-affinity BPs penetrated deeper than high-affinity ones. BPs were found to have high intensity in endocortical surfaces and haversian canals, whereas low intensity was found in canaliculi and osteocyte lacunae [162]. 

BPs are shown to bind plasma proteins in the bloodstream depending on pH, calcium, and species, including dogs, mice, etc. BPs rapidly disappear in circulation and mostly reach the bone due to their strong affinity to hydroxyapatite [163]. The liver is another organ where BPs are detected following oral and parenteral administration. However, to date, BPs have been described as transforming intracellularly to cytotoxic ATP analogs of non-nitrogen-containing BPs, which inhibits bone resorption. Renal clearance also affects drugs like BPs when reaching hard tissues, which correlates closely with creatinine clearance [157]. That is why patients whose creatinine clearance is below 30-35 mL/min are not approved for BP treatments [164]. Zoledronate is one of the BPs that are used for the treatment of bone resorptive diseases such as OP and osteoporotic fractures [165], and it has an adverse effect called acute kidney injury. In a study, 4405 patients who underwent zoledronate in five years were analyzed regarding creatinine levels after treatment. Both creatinine and glomerular filtration were significant factors in predicting acute kidney injury within 14 days; however, the recommended creatinine cut-off level (35 mL/min) showed a low sensitivity, but glomerular filtration was a good predictor. Therefore, the study suggested infusing zoledronate over 30 min in a patient with glomerular filtration less than 50 mL/min/1.73 m^2^ for the treatment of OP [164]. Another commonly used BP in bone targeting is alendronate, which is evaluated in many metastatic cancers (lung, breast, primary bone, and metastatic bone cancers) and bone-related diseases (OP) [166]. Modified nanoparticles with alendronate have been assessed for the treatment of OP regarding drug release behavior, cytotoxicity, and affinity to hydroxyapatite in vitro. These negatively charged nanoparticles showed sustained release of alendronate, good cytocompatibility, and blood biocompatibility with low hemolysis ratios (<5%), as well as exhibited a strong binding ratio to hydroxyapatite compared to nanoparticles with no alendronate [167]. Many drugs for bone treatment mainly target (1) bone resorption inhibitors, including BPs, estrogen and estrogen-related modulators, and calcitonin; (2) bone formation promoters like parathyroid hormone analogs; (3) vitamin K, vitamin D, etc. [168]. When taking the adverse effects (osteonecrosis of the jaw, synovitis, arthritis, etc.) of drugs into account, the development of new carriers is an urgent need. Wang et al. isolated extracellular vesicles from mouse MSCs and modified them with alendronate (Ale-EVs) for OP therapy. Modified Ale-EVs were infused into ovariectomy-induced OP rats to confirm their anti-OP effects. In vitro data showed a high binding affinity to hydroxyapatite. Compared to EV alone, Ale-EVs induced strong fluorescence in bone. Concerning the treatment point, Ale-EVs induced the growth and differentiation of mouse MSCs and ameliorated OP in ovariectomy-induced osteoporotic rats without any side effects [168].

### 3.2. Tetracyclines

Tetracyclines (TCs) are a class of broad-spectrum antibiotics targeting Gram-negative/positive bacteria and other intracellular bacteria such as *Rickettsia* and *Chlamydia*. However, recognizing their binding affinity to bone with a high turnover rate provided a new bone targeting strategy. The mechanism of action has shown that TC chelates the bond between oxygen atoms of TC and calcium of hydroxyapatite [169]. There are several types of tetracyclines based on their chemical structures and clinical uses: tetracycline, minocycline, doxycycline, tigecycline, omadacycline, eravacycline, and sarecycline. Several studies showed the bone-targeting properties of TCs in mice, rats, guinea pigs, rabbits, and dogs, in which a high fluorescence was detected in long and flat bones. Furthermore, administration sites of tetracycline and gender did not affect TC-induced bone fluorescence in both animal models and human trials [170]. TCs are evaluated in many bone-related diseases under several translational and clinical conditions. To treat osteoarticular tuberculosis, TC-modified nanoparticles were developed to deliver the anti-tubercular drug (rifapentine) into bone. After producing and confirming the effectiveness of TC nanoparticles in vitro, mice were intravenously injected with these nanoparticles to assess pharmacokinetics and biodistribution. The release of rifapentine from TC nanoparticles took more than 60 h, and 68% of TC nanoparticles could bind the hydroxyapatite in vitro. The drug concentration in bone from mice treated with TC nanoparticles was significantly higher than in control groups with no toxicity. 

As a result, rifapentine-loaded TC nanoparticles are a potential treatment that increases the drug concentration in bone and anti-tuberculosis activity, sustaining drug release and reducing dose [171]. TC-grafted mPEG-PLGA micelles were used to deliver astragaloside IV (AS) for osteoporotic improvement in rats. Besides their high uptake and fewer cytotoxic effects, they showed high binding affinity to hydroxyapatite. The ovariectomized female rats were injected with TC-mPEG-PLGA and mPEG-PLGA alone; TC-mPEG-PLGA was accumulated more in bone than mPEG-PLGA. Furthermore, the histopathological analysis indicated a thickened epiphyses line, regularly arranged trabecular bone with significance, and consistent bone mineral density value compared to untreated groups. Therefore, TC-mPEG-PLGA enhanced the anti-osteoporotic effect of AS in bone. In conclusion, TC-mPEG-PLGA promoted the absorption of AS due to the high diffusion and permeability provided by TC [172]. TCs (yellow fluorescent) have also been used in *Streptococcus mutans (S. mutans)*-induced bone loss as a fluorescent reagent in the measurement of bone formation activity in mice as compared to alizarin (red fluorescent)-injected and calcein (green fluorescent)-injected control groups [173]. The measurement of bone formation activity is exploited by a fluorescent reagent deposited on the calcification front, and the distance between two fluorescent labels of TC is determined at different time points (bone histomorphometry analysis) [174]. *S. mutans,* a TC-sensitive oral cavity bacterium, was intravenously inoculated into the mouse tail vein, followed by TC injections. MicroCT results showed a significant and apparent reduction following bacterial inoculation in bone mineral density of distal femurs regardless of injected fluorescent reagent. No significant differences existed between the TC–alizarin double-injected mice and the calcein–alizarin double-injected mice. One-time injection of TC neither affected bone microarchitecture nor ameliorated the alterations in bone mineral density in the *S. mutans*-induced bone loss model. Hence, TCs could potentially measure bone formation activity in the same way as non-antibiotic fluorescent reagents [173]. Furthermore, it was demonstrated that the administration of TCs inhibited osteoclastic activities by inhibiting MMP-9-mediated histone H3 cleavages, which alleviated bone resorption and induced the differentiation of osteoblasts. MMPs are inhibited through chelating Ca^2+^ and Zn^2+^. Acid production and secretion of lysosomal cysteine proteases are also inhibited by TCs [175,176]. Although TCs have been shown to have excellent properties for bone-targeting applications as guides for nanocarriers or drugs, their low chemical stability in different conjugation strategies may result in less optimal ligands for bone delivery [159]. Additionally, TCs were shown to promote osteogenic differentiation through different mechanisms. Tetracycline and doxycycline function under Wnt signaling while minocycline and sarecycline were found to be closely associated with Hedgehog signaling [177]. On the other hand, TCs inhibit osteoclast differentiation by suppressing MMP-9-mediated histone H3 cleavage through tigecycline and minocycline, which are potent inhibitors of MMP-9. Also, the administration of these TCs showed better outcomes in osteoporotic mice.

### 3.3. Acidic Amino Acid Oligopeptides

Acidic oligopeptides contain aspartic acid (Asp) or glutamic acid (Glu) residues in which the negative charge and polarity of side groups provide the binding affinity to calcium ions. Moreover, this affinity changed based on the type of polymeric amide bond and the number of amino acid residues. As the polymer chain becomes longer, the affinity and disassociation constant (Kd) increase, which enhances the binding affinity. Nevertheless, conjugating these peptides to the vectors requires some regulations: the size of the vector to be absorbed, ~6–8 repeated amino acids, the spatial configurations, and side chains [159]. As stated above, osteopontin and bone sialoproteins binding to hydroxyapatite include a repeated sequence of Asp and Glu. The number of residues of these amino acids exploits the affinity of molecules to hydroxyapatite. Even though isoform L-peptides or D-peptides do not determine this affinity, D-peptides provide advantages due to their compatibility with bone and low immune system recognition. A study stated that the hemopexin-like domain is a preferential site of Ca^2+^ binding, and hydroxyapatite has this site to bind these ions with four Asp carboxylate groups. Furthermore, calcium-binding was emphasized and mediated by carboxylate oxygens and carboxylate-rich matrix proteins (OCN, OPN, collagen, and tricarboxylate molecule citrate). All these molecules selectively bind to hydroxyapatite and increase the bone volume. In this regard, vitronectin was shown to bind both soluble Ca^2+^ and hydroxyapatite with high affinity and chemical selectivity. The average Kd was 27 ± 1.5uM [178]. To date, these peptides have been used for the delivery of enzymes, hormones, antibiotics, small interfering RNA (siRNA), and gene therapy.

In conclusion, the addition of high-affinity molecules to hydroxyapatite should improve current delivery and targeting strategies. Targeting newly formed hydroxyapatite and releasing drug/viral agents in such regions may provide new alternatives to access inside cells. To succeed, it is critical to find a better route to infuse such drugs at the beginning of development. 

### 3.4. Aptamers

Aptamers are short single-stranded nucleic acid (20–100 bases) molecules that interact with various proteins, cells, and small molecules, which can be used alone or multivalently [179,180]. Aptamers show strong chemical stability and can be rapidly produced on a large scale, and a variety of aptamer combinations can be made [179]. Furthermore, multivalent interactions of these oligonucleotides utilize viral entry, cell–cell and host–pathogen interactions, etc. Thus, molecular recognition-based applications of aptamers are highlighted as a promising targeted drug delivery method with a cost-effective and accessible perspective [181]. Compared to siRNA and oligonucleotide concepts, aptamers bind with high affinity and specificity to the protein targets. The binding sites of aptamers contain clefts and grooves of targeted molecules, which provide them with antagonistic activity. These agents show structural stability under different temperatures and storage conditions and can be chemically synthesized. Their strong chemical stability and rapid production give them several advantages over traditional approaches such as antibodies. Furthermore, aptamers can bind in many ways, which is superior to antibodies [179]. Increased aptamer concentrations in local administrations trigger higher interactions of aptamers and cell surface ligands. The binding of an aptamer to its ligand increases the affinity of other nearby aptamers to those specific targets. However, nucleases rapidly cleave these oligonucleotides, particularly in monomeric form [180]. Skeletal diseases are major clinical challenges to overcome due to various factors discussed in this review. Aptamers, in this regard, were found to be a critical mediator of targeted therapies since they can be attached to nanoparticles [182], viral vectors [183], exosomes from BM-derived MSCs [184], and proteins [185]. An osteoblast-binding aptamer, CH6, attached to lipid nanoparticles encapsulating Plekho1 siRNA (CH6-LNP-siRNA) targeting casein kinase-2 interacting protein-1 (CKIP-1) and increased cellular uptake and bone distribution compared to LNP-siRNA in osteopenic rodents. Here, targeting CKIP-1, a negative regulator of bone formation, is triggered only to form new bone, not bone resorption. CH6-functionalized lipid nanoparticles were taken up through micropinocytosis, and CH6 aptamers provided osteoblast-selective targeting in an in vitro setting [186]. Additionally, DOTAP-based cationic nanoparticles attached with (AspSerSer)_6_ delivered osteogenic siRNA (Plekho1) specifically to the bone-forming areas in osteoporotic rats. Along with forming new bone after in vivo systemic delivery, rats also showed enhanced bone micro-architecture and increased bone mass [187]. Exosomes were shown to balance bone metabolism and remarkably led BMSCs to differentiate into osteogenic cells in vitro. Administration of BMSC-derived exosomes (STExos) to OVX-induced postmenopausal osteoporotic mice resulted in poor amelioration. Following this insufficiency, BMSC-derived exosomes were functionalized with specific aptamers (STExos-aptamer complex). Intravenous injection of the STExo-aptamer complex increased bone mass in OVX mice as compared to STExo alone [188]. A similar approach has been proposed in the transplantation of synovial fluid (SF)-derived MSCs (SF-MSCs) for the treatment of OA [184]. In the study, kartogenin, a small molecule inducing differentiation of SF-MSCs to chondrocytes, was delivered to SF-MSCs via engineered exosomes, resulting in strong promotion of chondrogenesis of SF-MSCs both in vitro and in vivo. Moreover, the SF-MSC-targeting ability of these exosomes was enhanced by adding the MSC-binding peptide E7 (E7-Exo) to deliver kartogenin efficiently. As a result, E7-Exo transporting kartogenin easily accessed the cells and induced the differentiation of cartilage cells more than kartogenin alone or kartogenin-carrying exosomes with no modification in OA rats [184]. Gene delivery by viral vectors shows remarkable advancements in many diseases. However, some vectors, such as AAVs, represent a broad tissue tropism, leading to ineffective treatment in particular tissues. Puzzo et al. were able to chemically modify the AAV capsid surface by substituting amino acids without disrupting their packaging efficiency, and these novel AAVs were named Nε-AAVs. Nε-AAVs were individually functionalized via conjugating with DNA (AS1411), RNA (E3) aptamers, or a folic acid residue (FA). These modifications increased AAV transduction approximately 3- to 9-fold as compared to non-conjugated counterparts in vitro. Aptamer-mediated AAVs were administered into tumor cells in transplanted mice. The results showed that all three AAVs infected the tumor cells and reached the highest expression by day 14 without an off-target effect in peripheral organs [183]. In different preclinical studies, novel aptamer-related approaches have been implemented. For example, sclerostin loop3-targeting aptamer (Apc001PE) accelerated bone formation in osteogenesis imperfecta (OI) mice without any cardiovascular risk, which is superior to OI antibodies, causing a potential heart attack, stroke, and cardiovascular death [185]. In another study, a nanomedicine called DNAM or DNA-MSN containing an anti-sclerostin DNA aptamer (Aptscl56) layer and PEGylated dendritic mesoporous silica nanoparticles has effectively normalized osteoporotic bone loss in OVX mice, with bone enrichment, reduction in sclerostin levels, improvement in bone histomorphology parameters, and recovery in plasma bone turnover markers. In other nanoparticle-related bone targeting strategies, the mechanism of action of DNAM-immobilized Aptscl56 depends on the chelating activity and interaction between phosphate groups of the Aptscl56 and calcium in hydroxyapatites [182]. 

## 4. Future Perspectives

In this review, apart from a comprehensive understanding of bone and cartilage components and their mechanisms, we also highlighted potential molecular targets in bone and cartilage formation. Undoubtedly, inorganic components for bone targeting can be extensively modified, and this will take place more in the future. However, several drawbacks to reaching the targeted tissue, especially cartilage and bone, limit these drugs when correcting disease progression. Therefore, well-designed therapies are required. Organic components are highlighted targets, particularly in bone, and recently, receptor-mediated treatment has gained attention in gene therapy. Receptor mediation among these approaches is a rising star in target bone under various therapies, including direct infusion or cell therapies. Our special targets are chondrocytes that are highly affected by the mutations in the *GALNS* gene, resulting in MPS IVA with skeletal dysplasia (Figure 6). By knowing that cartilage and bone are hard-to-treat tissues due to the lack of blood vessels, we believe that (1) a consideration of receptors such as collagen receptors, BMPs, GPCRs, EGFR, VEGFR, or other small signaling molecules playing critical roles in chondrogenic development, (2) regulation of downstream pathways such as non-canonical Wnt signaling or Ihh signaling in early development, and (3) regulating ion channels may provide better targeting efficiency of desired drugs. For example, the delivery of anti-β-secretase (BACE1) or iduronate-2-sulfatase (IDS) into the central nervous system surrounded by the blood–brain barrier has been successfully implemented through transferrin receptor-mediated transcytosis. These studies should be considered as a tremendous achievement in difficult-to-reach tissue targeting. With this perspective, a deep understanding of cartilage and bone delivery may provide the appropriate approach. The preclinical gene therapy approaches indicate the importance of targeted tissue delivery by utilizing receptor mediation because fewer administered agents reach the target area, like growth plates of long bones. Therefore, utilizing secondary mechanisms, including receptor-specific tags on enzymes or transgenes, robust expression of the transgene in bone marrow and liver as a potential factory of desired enzyme production, and secretion locally and into circulation might be some of the alternatives in the current state of the art. Unfortunately, there are still many unknown mechanisms underlying cartilage and bone targeting, and more investigation is required to understand how these cells in the growth plate and deep zones of articular cartilage maintain their presence without blood supplies. Understanding these mechanisms will contribute to developing new therapeutic strategies in bone metabolic disease.

## Figures and Tables

**Figure 1 ijms-25-08339-f001:**
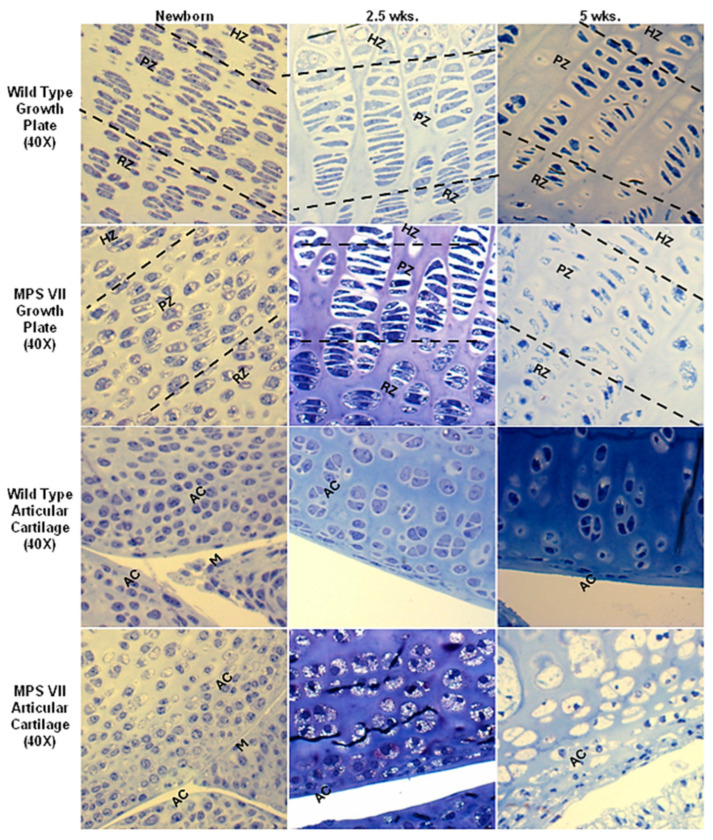
Cartilage of MPS VII vs. wild type (WT) mouse model at an early stage of life. Toluidine blue staining of the knee joint from 2–3-day-old (**left**), 2.5-week (**middle**), and 5-week-old (**right**) MPS VII mice at 40X magnification. Top horizontal panels: Growth plate of MPS VII and WT mice (HZ: Hypertrophic zone, PZ: Proliferative zone, RZ: Resting zone). Bottom horizontal panels: Articular cartilage of MPS VII and WT mice (AC: Articular cartilage, M: Meniscus). Wild-type femur growth plate shows more transparent chondrocytes, particularly in the resting and proliferation zone, than MPS VII (a deficiency of β-glucuronidase (GUS^−/−^) causing glycosaminoglycan accumulation in multiple tissues). Furthermore, healthy chondrocytes create smooth column structures and show high proliferation rates. In MPS VII, clear vacuolation is observable at birth and increases in number with age. Furthermore, the disorganization of columnar structures is progressive in AC. Adapted from [12].

**Figure 2 ijms-25-08339-f002:**
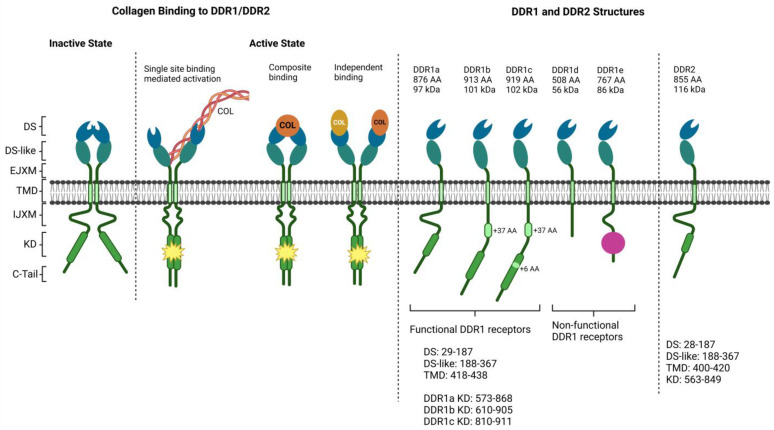
Structures and types of DDR-1 and -2. DDR1d and -e represent non-functional DDR-1 receptors [31,32,33]. This figure was created with BioRender.com.

**Figure 4 ijms-25-08339-f004:**
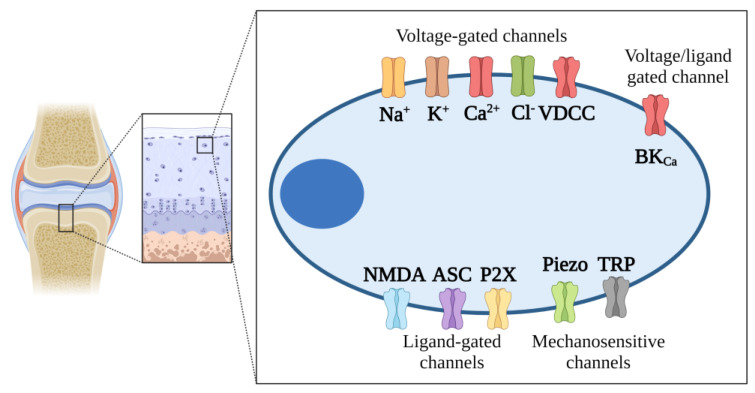
Chondrocyte plasma membrane-attached ion channels. The figure shows a schematic representation of the most common channels identified in states of health and diseases in chondrocytes. VDCC: Voltage-dependent calcium channel. BK_Ca_: Ca^2+^-activated K^+^ channel. NMDA: N-methyl-d-aspartate. ASC: Acid-sensitive channel. P2X: Purinergic receptor. This figure was created with BioRender.com.

**Figure 6 ijms-25-08339-f006:**
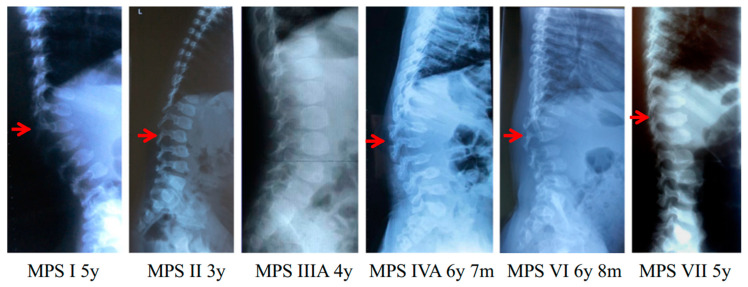
Dorsolumbar spine X-ray pictures from patients affected by MPSs I, II, IIIA, IVA, VI, VII at ages 3 to 7. All patients shown here were diagnosed with a severe type of each MPS. MPS I has a high lumbar kyphosis at L2 and less extent around this vertebra. MPS II with the inferior beaking of ovoid vertebrae at the lumbar, widening of intrapediculate space; however, the progression of bone deformity is mild and slower. MPS III has an ovoid vertebrae at the lumbar and mild widening at interpediculate spaces. In MPS III, the progression rate of bone deformity is the mildest and slower among all MPSs. MPS IVA with prominent lumbar kyphosis at L2. Platyspondyly is a universal sign in a central anterior beaking compared to the inferior. In the most severe ones, interpediculate spaces notably increased, multiple vertebral subluxations and small sacrum are clearly seen. MPS IVA has the most severe skeletal deformity among all MPS types and low bone mineral density. MPS VI with prominent kyphosis at L2, universal platyspondyly similar to MPS IVA, the moderate posterior scalloping of the lumbar vertebrae, hypoplastic lumbar vertebrae (L1–L5) with a characteristic superior notch (L4 and L5), multiple vertebral subluxations, a marked increase in interpediculate spaces, small sacrum, and low bone mineral density. MPS VII with kyphosis at L1, moderate ovoid vertebrae at lumbar (distinguishing from abnormality of MPS IVA), the lumbar vertebrae with widening of interpediculate spaces, universal platyspondyly of dorsolumbar vertebrae with a mild central anterior beaking. MPS VII has less severity compared to MPS IVA regarding the radiological screening. Adapted from [12]. Red arrows: L2 vertebra with kyphosis.

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
