# Peer review of "Potential Targeting Mechanisms for Bone-Directed Therapies"

_ijms, 2024, doi:10.3390/ijms25158339_

Round 1

Reviewer 1 Report

Comments and Suggestions for Authors

Comments to the Authors of manuscript number ijms-3044102 entitled “Potential Targeting Mechanisms for Bone-Directed Therapies

Bone development involves complex mechanisms and any misregulation can lead to skeletal dysplasia. Due to the avascularity of cartilage and bone, drug delivery is challenging. Recent novel bone-targeting approaches show promise but do not fully correct skeletal dysfunctions, especially in growth plates. Traditional therapies partially help, but new methods are needed. This review covers current strategies for bone targeting and highlights potential advances in drug delivery and bone development regulation for future clinical use.

1. L 35- abbr. explain needed

2. L 53-  Inorganic phosphate add

3. L 55- dot should be omitted

4. L 57- VEGF- full name needed

5. L 58- what factors?

6. L 60- osteoblasts secrete minerals?

7. L 62-hypertrophic cavities?

8. L 64- ECM should be explained above

9. L 63- 73- the text is chaotic and unclear.

10.L 75- “each zone” of what?

11. what is the hypothesis?

12. the lack of part of Material and methods. This section should include all the bases and databases searched, as well as the criteria used for selecting and analyzing data.

13. L 117- more is needed

14. 2.1.1 - The text repeatedly mentions the role of DDR-1 and DDR-2 receptors without adding new insights. Sentences are often long and complex, making it difficult to follow the key points. The text does not adequately explain how these pathways interact with DDR signaling to influence bone development and repair. The results from KO mice and other studies are mentioned without sufficient context or explanation of their significance. Text frequently jumps between different topics. Many full terms without clear definitions.

15. part 2.1.2.- the role of GAGs in bone formation is mentioned multiple times without expanding on the different mechanisms. A few Sentences too long. the text mentions the role of GAGs in inhibiting hydroxyapatite formation and interacting with collagen fibrils, it does not explain how these processes occur at a molecular level. There are several typographical errors and awkward phrasings, such as "suchs a suppressor" instead of "such as a suppressor".

16. L 224-227- The sentence is lengthy

17. L 228-230- More detail needed on how biglycan and decorin influence bone matrix deposition and mineralization.

18. 231-234- different species are mentioned- what exactly? If avian bone is specific so what is the difference as compared to other species and what species?

19. L 264-275- This section requires simplification

20. L 277-284- Briefly explain the significance of each glycoprotein listed.

21. L 285-320- The description of ALP's roles and mechanisms is repetitive.

22. L 321-334- More explanation on how ON specifically affects bone remodeling and mineralization is needed

23. L 335-349- how tetranectin's expression in osteoblasts affects bone mineralization.

24. L 351-359- specific diseases are needed where these glycoproteins are implicated.

25. L 360-366 – what is the connection between their role with clinical and pathological implications?

26. L 374-390- Specify the clinical implications of VnP-16 and its potential as a therapeutic agent.

27. L 391-411- what is the clinical relevance of OPN in various diseases?

28. L 453-462- what is the role of carboxylated and non-carboxylated forms of OCN?

29. please indicated clearly their role in bone metabolism with metabolic and clinical implications

30. L 463-473- what is the mechanism by which OCN and OPN modulate MSC function?

31. L 474-485- what are clinical implications of MPG deficiency?

32. 2.1.7.1 - Connect integrin expression patterns to pathological conditions like OA for clinical context. Provide specific examples or mechanisms of how ion channels regulate chondrocyte homeostasis. Highlight the clinical implications of Piezo channels in diseases like RA and osteoporosis.

33. 2.1.7.3. Please provide a succinct explanation of GPCR structure and its interaction with G proteins. Connect GPCR-mediated signaling to specific physiological processes in chondrocytes. Discuss the potential of targeting specific GPCRs for therapeutic interventions in chondrocyte-related diseases.

34. 2.1.7.4. Discuss the clinical implications of Wnt pathway dysregulation in diseases like OPPG and sclerosing diseases related to SOST deficiency. Discuss the impact of inflammatory cytokines like IL-1β on LRP5 expression and its downstream effects in chondrocyte pathology. Explain the role of LRP-1 in regulating ADAMTS-5 and MMP-13 activity in the context of OA pathology.

35. 2.2. - a detailed description of bisphosphonates (BPs) as therapeutic agents for bone-related diseases is needed. Explain how BPs bind to hydroxyapatite and their mode of action in inhibiting osteoclast activity. What is the cellular mechanism of BPs? What is the distribution pattern of BPs in cortical and trabecular bone and their detection using bone scanning techniques?

36. 2.2.2. an overview of tetracyclines as antibiotics and their newfound role in bone targeting due to their affinity for hydroxyapatite is needed. Explain how tetracyclines bind to hydroxyapatite, facilitating targeted drug delivery to bone tissues. What is the mechanisms by which tetracyclines inhibit osteoclastic activities and promote bone formation?

37. 2.2.3. how the length of the polymeric chain and the number of amino acid residues influence the affinity and disassociation constant (Kd) for calcium ions and hydroxyapatite?

38.2.2.4. what is aptamers' role in targeted drug delivery? specific examples of aptamer applications in bone-related therapies, including enhancing osteogenic siRNA delivery, promoting bone formation, and normalizing bone loss in osteoporotic models are needed. The comparison between aptamer-based therapies and conventional approaches is needed. What are future directions?

Reviewer 2 Report

Comments and Suggestions for Authors

In the introduction, the authors review the cells involved in bone formation as well as the metabolic pathways involved in cell differentiation, establishing the basis for the subsequent development of the review work. It is a narrative review on basic sciences that tries to establish elements for the resolution of clinical problems.

The bibliographic strategy followed should be commented on more extensively to allow other groups to carry out work of similar characteristics.

This is followed by a description of the protein elements of bone with special reference to collagen. Special reference is made to the mechanism of collagen binding with a graphical representation that helps to understand this aspect. The role of other proteins involved in the bone remodeling process is then described.

The role of surface membranes is described and the graphical representation of membrane channels can be observed. The role of drugs ( BPs ) and diagnostic procedures ( tetracyclines ) is introduced.

It would be interesting to refer to the clinical significance of the described mechanisms and the repercussions on patient management.

The bibliography is extensive and up to date
